# SARS-CoV-2 Clinical Outcome in Domestic and Wild Cats: A Systematic Review

**DOI:** 10.3390/ani11072056

**Published:** 2021-07-09

**Authors:** Sebastian Giraldo-Ramirez, Santiago Rendon-Marin, Javier A. Jaimes, Marlen Martinez-Gutierrez, Julian Ruiz-Saenz

**Affiliations:** 1Grupo de Investigación en Ciencias Animales—GRICA, Facultad de Medicina Veterinaria y Zootecnia, Universidad Cooperativa de Colombia, Bucaramanga 680002, Colombia; sebastian.giraldor2@udea.edu.co (S.G.-R.); santiago.rendonm@udea.edu.co (S.R.-M.); marlen.martinezg@campusucc.edu.co (M.M.-G.); 2Department of Microbiology and Immunology, College of Veterinary Medicine, Cornell University, Ithaca, NY 14853, USA; jaj246@cornell.edu; 3Infettare, Facultad de Medicina, Universidad Cooperativa de Colombia, Medellín 050012, Colombia

**Keywords:** cats, coronavirus, felids, virus, SARS-CoV-2

## Abstract

**Simple Summary:**

In view of the current global situation due to the COVID-19 pandemic and considering the evidence that SARS-CoV-2 has the ability to infect multiple species, this work was aimed at identifying the clinical signs of SARS-CoV-2 infection in domestic and wild felids. Our results evince that the signs developed in felids show similar progression to those occurring in humans, suggesting a relationship between the viral cycle and target tissues of the virus in different species.

**Abstract:**

Recently, it has been proved that SARS-CoV-2 has the ability to infect multiple species. This work was aimed at identifying the clinical signs of SARS-CoV-2 infection in domestic and wild felids. A PRISMA-based systematic review was performed on case reports on domestic and wild cats, reports on experimental infections, case reports in databases, preprints and published press releases. Descriptive statistical analysis of the data was performed. A total of 256 articles, 63 detailed official reports and 2 press articles on SARS-CoV-2 infection in domestic and wild cats were analyzed, of which 19 articles and 65 reports were finally included. In domestic cats, most cats’ infections are likely to be asymptomatic, and 46% of the reported infected animals were symptomatic and predominantly presented respiratory signs such as sneezing and coughing. In wild felines, respiratory clinical signs were most frequent, and up to 96.5% of the reported affected animals presented coughing. It is noteworthy that, to date, symptomatic animals with SARS-CoV-2 infection have been reported to belong to two different subfamilies (*Phanterinae* and *Felinae*), with up to five different felid species affected within the *Felidae* family. Reported results evince that the signs developed in felids show similar progression to those occurring in humans, suggesting a relationship between the viral cycle and target tissues of the virus in different species. While viral transmission to humans in contact with animal populations has not been reported, spill-back could result in the emergence of immune-escape mutants that might pose a risk to public health. Despite the clear results in the identification of the typical clinical picture of SARS-CoV-2 infection in felines, the number of detailed academic reports and papers on the subject is scarce. Therefore, further description of these cases will allow for more accurate and statistically robust clinical approaches in the future.

## 1. Introduction

Coronaviruses are infectious agents that are globally widespread and affect different animal species, including humans, and jumping the species barrier is well known in the transmission of these pathogens [1]. Since the severe acute respiratory syndrome coronavirus type 2 (SARS-CoV-2), which causes coronavirus disease 2019 (COVID-19) [2], first became widely known at the end of 2019, the eyes of the world have been focused on this infectious agent in search of an understanding of its origin, replication cycle, transmission and evolution. As of 31 May 2021, COVID-19 has been responsible for more than 3.5 million deaths and has been identified in most countries of the world [3].

SARS-CoV-2 is an enveloped virus with single-stranded unsegmented ribonucleic acid (RNA). It belongs to the order *Nidovirales*, family *Coronaviridae*, subfamily *Orthocoronavirinae* and genus *Betacoronavirus* [4]. SARS-CoV-2 has been described as a zoonotic virus and possibly emerged from a betacoronavirus originally from bats [5], with an unknown intermediate host. Proposed intermediate hosts include pangolins, rodents, and bats, but the transmission dynamics before jumping the species barrier to humans remains unclear [6,7,8,9].

Considering the wide range of hosts that coronaviruses can infect, studies have been conducted during 2020 and through 2021 to determine the affinity of angiotensin-converting enzyme 2 (ACE2) from different animal species for the SARS-CoV-2 spike protein (S) [10]. It has been shown that susceptibility to infection in companion animals varies depending on the species affected. In cats and members of the family Felidae, susceptibility to infection is higher due to the high affinity of the virus to the ACE2 orthologue [11]. In contrast, there seems to be a lower risk of infection and transmission in dogs due to the low amount of ACE2 expressed in the respiratory tract cells, even though it has been suggested that SARS-CoV-2 may enter the cells using this or other receptors [12,13].

Coronaviruses affecting the family *Felidae* have been comprehensively described. Different viruses from the family *Coronaviridae* are capable of infecting domestic and wild felines [14]. Classified in general as feline coronaviruses (FCoVs), some viruses usually cause mild enteric disease (feline enteric coronaviruses; FECV). However, these viruses can undergo internal mutation processes and evolve into feline infectious peritonitis virus (FIPV), which has increased virulence and causes a severe disease with fatal outcomes [15,16].

Although SARS-CoV-2 is different from the FCoVs widely distributed in both wild and domestic feline populations (the former belonging to the genus *Betacoronavirus* and the latter to the *Alphacoronavirus*) [17], SARS-CoV-2 infections reported in felines have been associated with studies focused on experimental infections and transmission from humans. In these studies, both infection and cat–cat transmission have been demonstrated, and the virus has been found in nasal and pharyngeal swab samples within 2 days post infection. Viral RNA has been detected in tissue samples such as nasal turbinates, soft palate, tonsils, trachea and lungs, proving the susceptibility and spread of SARS-CoV-2 in cats. Furthermore, viral RNA can be detected in the respiratory tract tissues of non-infected cats that have been placed together with experimentally infected cats [18,19,20]. Thus far, no antibody cross-reaction between FCoVs and SARS-CoV-2 has been reported [21].

In view of the increasing number of reports of felines infected with SARS-CoV-2 and the close relationship between owners and domestic cats and between zoo or wildlife center keepers and wild felines, there is a need to understand the clinical implications of SARS-CoV-2 infections in felines. In this review, we analyzed the available reports regarding clinical signs in felines diagnosed with SARS-CoV-2 worldwide, aiming to provide an overview of the typical clinical picture of the disease in these species. Further, we sought to identify and compile the available information from experimental infection studies in felines, as well as clinical cases in wild felines and the associated clinical signs.

## 2. Materials and Methods

### 2.1. Data Collection

Considering the cohabitation of humans and animals in limited spaces and the proven susceptibility of certain animal species (such as felines) to SARS-CoV-2, a retrospective study was performed to systematically review articles on case presentations and reports of SARS-CoV-2 infections in domestic and wild cats worldwide, according to the international principles of preferred reporting items for systematic reviews and meta-analyses (PRISMA) [22] statement. A systematic literature review was performed, which included published and preprint papers that reported results of natural and experimental infections in wild and domestic felines. This study is registered with PROSPERO, CRD42021255852.

The reviewed articles were retrieved from the databases of the National Center for Biotechnology Information at the National Library of Medicine (MedLine/Pubmed; http://www.ncbi.nlm.nih.gov/pubmed); Scientific Electronic Library Online (SciELO; https://scielo.org/); Red de Revistas Científicas de América Latina y el Caribe, España y Portugal (Redalyc; https://www.redalyc.org/); Scopus (https://www.scopus.com/) and Biorxiv (https://www.biorxiv.org/). The accession date for all databases was 15 March 2021. In addition, reports published by the World Organisation for Animal Health (OIE) and the United States Department of Agriculture (USDA) were retrieved as official sources of information to obtain updated data on a weekly basis.

To systematize the information, a database was created. It included the references of the selected articles, name of the article, authors, year of publication, country or countries of study, countries involved in the research, type of infection (natural or experimental), percentage of positive samples found, signs observed, relevant findings and species.

### 2.2. Search Strategy

Search, selection and collection of the articles from the databases were conducted using keywords from Medical Subject Headings database and Boolean connectors. A general search was performed, followed by a more detailed search for scientific publications associated with the different official reports found. All review and opinion articles on SARS-CoV-2 infection in felines, as well as articles concerning serological diagnosis or referring to other animal species, were manually excluded.

In all databases, the following search strategies were used: “SARS-CoV-2 AND cats AND clinical signs”; “SARS-CoV-2 AND cats AND symptoms”; “SARS-CoV-2 AND cats AND infection”; “SARS-CoV-2 AND experimental infection AND cats”; “SARS-CoV-2 AND lions AND clinical signs”; “SARS-CoV-2 AND tigers AND clinical signs”

Inclusion criteria: Case reports on domestic cats and wild felines, experimental infections, case reports in databases of OIE, USDA and the Centers for Disease Control and Prevention, preprints and press releases. The time period was from 1 January 2020 to 15 March 2021.

Exclusion criteria: Reviews, seroprevalence studies, cases in non-feline species and reports related to infections in humans or vaccination.

A two-phase approach was used for the selection of articles. During the first phase, articles were selected based on titles and abstracts, excluding those considered irrelevant to the topic. In the second phase, the entire text of each article included in the first phase was reviewed. Articles that did not meet the pre-established criteria were excluded. To achieve methodological robustness, article selection and data cleansing were performed independently by two researchers and all disagreements were resolved by discussion with the research team, as previously reported [23].

### 2.3. Quality Assessment

The selected publications were evaluated by the criteria of the GRADE Guide (Grading of Recommendations Assessment, Development and Evaluation) [24,25], which classifies the quality of evidence in one of four levels—high, moderate, low, and very low. The assessment of the included studies was conducted in a single-blind manner by two researchers.

### 2.4. Statistical Analysis

Descriptive statistics was used in all cases to evaluate the results (frequencies and percentages). Statistical analyses were carried out using Excel^®^ for Windows^®^ (Redmond, WA, USA).

## 3. Results

After applying the different search strategies in the databases and screening the information to identify cases associated with felids diagnosed with SARS-CoV-2 infections and the reported clinical signs, a total of 19 articles and 65 detailed reports from official sources were included (Figure 1).

The quality of the selected publications was estimated to be “moderate” to “low quality” due to the lack of epidemiological studies of the impact and distribution of SARS-CoV-2 in cats at a population level. Otherwise, quality assessment in official reports by USDA/OIE present a “Very low quality” according to the GRADE classification. The OIE website was the main official source of information, which had some reports that were already published in indexed journals and preprint servers. It is noteworthy that, within the time period of the present article, 76 confirmed cases of SARS-CoV-2 in domestic cats have been reported by the USDA [26]. However, these reports do not provide any information on the diagnosis or diagnostic methods used. Therefore, based on whether they provide relevant information regarding the occurrence or absence of clinical signs, only 22 out of the 76 cases in the United States were included in the analysis.

The distribution of cases in felids shows a global pattern, with most cases reported in the Americas and Europe (Figure 2); this is consistent with the fact that these two regions have been the main foci of COVID-19 cases in humans during the course of the pandemic. However, feline infections with SARS-CoV-2 in other highly prevalent regions (e.g., India, Brazil and other Latin American countries) may be underestimated, as more studies are not yet available in these regions.

### 3.1. Clinical Presentation of SARS-CoV-2 Infection in Felines

#### 3.1.1. Domestic Cats

The first case of SARS-CoV-2 infection in a domestic cat was reported in Belgium at the beginning of the pandemic (March 2020) [27]. As of 15 March 2021, a total of 124 SARS-CoV-2 cases have been reported in cats as follows: 76 in the United States; 10 in the United Kingdom; 5 in Hong Kong; 4 each in Brazil, Switzerland and Argentina; 3 each in Chile and Canada; 2 each in Germany, Japan, France, Spain and Greece; and 1 each in Belgium, Russia, Italy, Estonia and Latvia [28] (Figure 2). Relevant information regarding the occurrence or absence of symptoms was obtained from these reports. The most common clinical signs associated with SARS-CoV-2 infection in domestic cats were summarized, which provided an outline of the typical clinical picture of the infection.

Out of the 124 SARS-CoV-2 cases in cats reported to date, 70 presented clear information on the occurrence or absence of clinical signs. In 54% of the cases (38 cats), the animals were asymptomatic. These cases were mainly identified because they were closely related to human patients diagnosed with COVID-19. The patients were usually the owners of the cats. These cats, in turn, were screened by national or local research programs seeking to establish the presence of the virus in animals in relation to human cases [26,28].

Furthermore, 46% of the cases (32 animals) showed clinical signs which could be associated with SARS-CoV-2 infection; 26 of these animals recovered. Six animals died over the course of the infection, three of which were euthanized and three died as a consequence of medical complications. One of these cases (reported in the United States) showed respiratory signs such as shortness of breath and tachypnea. In addition, the cat was diagnosed with hypertrophic cardiomyopathy and *Mycoplasma haemofelis* infection [29]. Another case of death due to medical complications following SARS-CoV-2 diagnosis was also reported in the United States. In this case, a cat showed clinical signs associated with the respiratory tract, nasal discharge and neurological signs such as head pressing. The necropsy results suggested that the cat had bacterial meningoencephalitis [28].

Another case that resulted in the death of the infected animal was reported in Spain. The cat presented dyspnea and acute tachypnea that required veterinary care. The cat was diagnosed with anemia, thrombocytopenia and hypertrophic cardiomyopathy. Ultrasound examination revealed a bronchointerstitial pattern in the lung. The disease course of the animal was unfavorable and euthanasia was required. Subsequently, a necropsy was performed and tissue samples from the lung, trachea, nasal turbinates, lymph nodes, tonsils, bone marrow, heart, spleen, kidney, small and large intestine, among others, were collected for histological studies. Tissue analysis revealed signs of severe pulmonary edema, hemorrhage and congestion; pulmonary thrombosis of capillaries and small- and medium-sized blood vessels; hemorrhages in nasal turbinates; interstitial fibrosis of the inter-ventricular septum and left ventricle of the heart; hepatic lipidosis; splenic hematopoiesis; moderate membranoproliferative glomerulonephritis and focal adrenal hyperplasia [28,29].

Among the most common clinical signs observed in the affected animals are those involving the respiratory tract, such as coughing, sneezing, shortness of breath, rales in the lungs, increased respiratory rate, congestion and eye discharge. In addition, digestive signs such as vomiting, mouth ulcers and some unspecified signs classified as “mild digestive” signs, presumably diarrhea, were reported. Non-specific signs included lethargy, fever and lack of appetite. Finally, in certain specific cases, neurological signs (head pressing), cardiac signs such as hypertrophic cardiomyopathy, as well as anemia and thrombocytopenia were observed, which could not be directly linked to SARS-CoV-2 infection (Table 1).

Recently, the new SARS-CoV-2 British variant (B.1.1.7), whose transmissibility is higher in humans [30], has also been reported in companion animals. In particular, this variant has been identified in eight cats that presented heart abnormalities, such as congestive heart failure and ventricular arrhythmia, secondary to myocarditis. One of the diagnosed animals presented complications associated with heart disturbances, due to which euthanasia was required [31].

As described before, the typical clinical manifestation of SARS-CoV-2 infection in domestic felines is mainly a respiratory-type infection. According to two detailed reports on the clinical course of SARS-CoV-2 infection in domestic felines, the onset of clinical signs in the infected animals started 7 days [27] and 17 days [32] after the onset of clinical signs in the human contact in the respective cases; however, it is difficult to accurately determinate the infection date. In the clinical presentation of SARS-CoV-2 infection in cats, 40.6% of the animals presented sneezing, which may be accompanied by nasal discharge. Approximately 6% of the animals presented with tachypnea, 9% with ocular discharge and 12.5% with shortness of breath. With regard to digestive signs, diarrhea occurred only in a few cases (6.25%). In addition to these main signs, the animals presented non-specific clinical signs common to many infectious conditions such as lethargy, fever and lack of appetite. Hypertrophic cardiomyopathy was reported in 9% of the diagnosed animals. This clinical finding was linked to the death of two affected animals as it represents a major life-threatening clinical complication [28]. Similarly, other heart disturbances such as congestive heart failure (25%) and ventricular arrhythmia (12.5%) were also observed and related to the B.1.1.7 variant [31].

Although neurological signs and hematological alterations such as thrombocytopenia and anemia have been reported, these are more likely to be pre-existing conditions in the affected animals, which worsened the clinical condition resulting from SARS-CoV-2 infection [28].

Based on the information from some reports of symptomatic cases of SARS-CoV-2 infection in domestic cats, most of the animals usually recover in 7–15 days after the onset of symptoms [27,32,33]. This may vary depending on the particularities of each animal, external factors such as timely veterinary care and physiological factors such as age, sex, diet, reproductive stage and underlying diseases [20].

Due to the limited information concerning clinical signs of SARS-CoV-2 infection that has been described and reported to date, it is not possible to establish a robust statistical association supporting the information compiled here. The following table of SARS-CoV-2 in domestic felines is a preliminary overview of the signs commonly recorded in the reports.

#### 3.1.2. SARS-CoV-2 in Wild Felids

Although most case reports on SARS-CoV-2 infection in felids correspond to domestic cats, several cases in wild felines have been described in different places around the world [28]. Members of the genus *Panthera*, such as tigers, lions and snow leopards, as well as members of the genus *Puma*, such as *Puma concolor*, have been affected by SARS-CoV-2 (Figure 3) [40,41].

SARS-CoV-2 infection in these wild felines has mainly been associated with human–animal transmission, since the staff caring for these felines tested positive for COVID-19 and had probably transmitted the virus to the animals. In a case in the Bronx Zoo in New York, the sequences identified in keepers and tigers shared close phylogenetic relationships, and it was suggested that the transmission occurred from the former to the latter. Regarding lions, it was concluded that the virus was introduced again into the zoo, the mechanism of which is not entirely understood, since the virus identified in the keepers and the tigers belonged to a genotype different from the one found in the lions [42].

Despite the genotypic differences in the sequences found in the Bronx Zoo, all animals positive for SARS-CoV-2, except for one asymptomatic tiger, presented clinical signs associated with the upper respiratory tract, such as coughing and wheezing, but no breathing difficulties. Diagnosis was performed using fecal samples from all animals (tigers and lions) and additionally through nasal and oropharyngeal swabs, as well as tracheal wash from one symptomatic tiger. All samples were analyzed by quantitative reverse transcription polymerase chain reaction (RT-qPCR) [28]. However, the virus was not isolated.

A case of a puma diagnosed with SARS-CoV-2 infection by RT-qPCR in South Africa was also linked to interaction with a human positive for the virus as a possible source of infection. However, no clinical signs were observed in this animal. Another wild feline described as susceptible to the virus is the snow leopard (*Panthera uncia*). Three confirmed cases of SARS-CoV-2 infection in snow leopards showing respiratory signs, such as coughing and some wheezing, were reported in Kentucky, United States. All animals gradually recovered from the clinical signs [28].

To date, two cases of death by natural causes have been reported in wild felids as a result of worsening health status after being diagnosed with SARS-CoV-2 infection. In Angelina County, Texas, United States, a 20-year-old Malayan tiger was diagnosed with SARS-CoV-2 infection. The animal showed worsening respiratory signs and unsatisfactory treatment response and was eventually euthanized. In Sweden, a 17-year-old female tiger was euthanized after worsening of its clinical respiratory signs. Although there is no further detailed information on the clinical causes that resulted in the death of the animals, advanced age may have been a determining factor in the fatal outcome of these cases [28].

Considering that the development and clinical manifestation of SARS-CoV-2 infection may vary among species, in the case of tigers, lions and snow leopards, and even more among other subfamilies, in the case of pumas, the typical clinical picture of SARS-CoV-2 infection in wild felines is uniform. Among 29 wild felids reported as symptomatic after being diagnosed with SARS-CoV-2 infection, the clinical manifestation was mainly respiratory tract alterations. Sneezing and coughing were the most common clinical signs in wild felines, occurring in 23 and 28 animals, respectively. Together with respiratory signs, non-specific clinical signs, such as lack of appetite and lethargy, have been observed, with the former being identified in 15 animals out of the 29 wild felines reported. Other signs such as nasal discharge and wheezing have also been reported, but to a lesser extent (Table 2).

### 3.2. Experimental Infections

In addition to the case reports associated with natural infections worldwide, mainly resulting from human–cat contact, experimental infections have been performed in cats to determine their susceptibility to SARS-CoV-2 infection [12] and clinical and histopathological implications of the infection [20]. Human–cat transmission is highly probable; however, it is important to determine the likelihood of cat–cat transmission. Hence, a common objective in experimental infections has been to assess the possibility of transmission from experimentally infected animals to healthy animals [18,20,43].

Most experimental infection studies have used similar protocols, beginning with the viral challenge of healthy animals through intranasal inoculation of the virus to simulate one of the most probable viral entry routes into the respiratory tract. A total of 63 animals have been evaluated in experimental studies. Close-contact tests between experimentally infected cats and healthy cats were performed to investigate cat–cat virus transmission. The time lag between viral challenge of infected cats and exposure of healthy cats was typically 1 day [18,20,43]. The virus was detected in samples from nasal and rectal swabs [18,43] or in samples from biopsies of nasal turbinates, soft palate, tonsils, trachea, lungs, and small intestine in healthy animals exposed to infected animals.

According to the results of these experimental studies, most animals did not show clinical signs of infection, except for one experiment that described clinical signs such as diarrhea and weight loss, without any further complications [43]. Only one of the tested animals died, albeit the cause of death remains unclear [20].

In addition, it has been experimentally described that cats previously infected with SARS-CoV-2 can be re-infected with the virus. Viral RNA has been detected in nasal and oropharyngeal swabs, although no rectal shedding was reported. Moreover, the re-infected animals did not transmit the virus to healthy animals, even when in close contact, as no viral RNA shedding was detected in any of the analyzed samples [44].

Regarding histopathological findings, SARS-CoV-2 infection may cause lesions in the nasal and tracheal mucosa and lung parenchyma, which is associated with virus replication in these tissues [12]. The histopathological lesions, identified after the animals under study were euthanized, showed suppurative lymphoplasmacytic rhinitis in nasal turbinates, as well as lymphoplasmacytic tracheitis and alveolar histiocytosis. Additionally, moderate lung changes such as interstitial lymphocytic pneumonia with perivascular and peribronchial cuffing were observed. These findings were reported in experimentally infected animals after sacrifice, even in the absence of clinical signs of SARS-CoV-2 infection [20]. The association of the lesions observed in the histopathological analyses with the presence of the virus in these tissues was evinced by the detection of viral RNA using RT-qPCR and subsequent viral titration in cell culture [12].

### 3.3. Strengths and Weaknesses 

To the best of our knowledge, this work is the first collection of reported data related to SARS-CoV-2 infection in felids with clinical symptoms. This work was aimed at providing a structured view of the clinical picture of SARS-CoV-2 infection and a tool to understand the clinical implications of the infection in these animals. However, it is worth mentioning that as mentioned in the quality evaluation, the present work is limited by the fact that there is not enough described and uniformly presented information in the official reports that allows a more precise delineation of the clinical picture of SARS-CoV-2 infection in felids; this is because most of the existing reports lack detailed descriptions of the clinical manifestations of the cases. Due to the intrinsic objective of the study, no seroprevalence studies were collected, which leads us to have a strong bias of underestimating asymptomatic and mild cases.

## 4. Discussion

SARS-CoV-2 has been described as the etiological agent of COVID-19, a disease that is associated with a wide range of clinical signs mainly in the respiratory tract in humans, as commonly observed in infections with severe acute respiratory syndrome coronavirus (SARS-CoV) and Middle East respiratory syndrome coronavirus (MERS-CoV) [45]. As of late 2020, human patients diagnosed with SARS-CoV-2 have been reported to show signs primarily associated with the respiratory tract, and signs such as coughing and dyspnea, which occur in 76% and 55% of cases, respectively, are among the most frequent clinical manifestations of COVID-19 [46]. Likewise, non-specific clinical signs, such as fever, lethargy and headache, usually observed in most cases of infection, have been recorded as some of the most common signs in human patients, with fever in more than 90% of symptomatic COVID-19 cases [47,48]. Gastrointestinal signs, such as diarrhea, are also observed in human patients, albeit less frequently than those mentioned related to a respiratory infection. In addition, cardiac problems have been reported in up to 12% of cases, evincing that COVID-19 is not a disease limited to respiratory tissue lesions [46,48].

Despite the inherent differences between humans and felids, the clinical characteristics of disease progression following SARS-CoV-2 infection are similar between both. Confirmed cases of SARS-CoV-2 infection in domestic cats have been steadily reported in multiple countries ever since the virus became globally prevalent. Such cases were mainly related to COVID-19 cases in humans and owner–companion transmission due to close contact within households [17,28]. Moreover, based on evidence from experimental studies, other SARS-CoV-2 cases in domestic cats could be ascribed to transmission by direct cat–cat contact [12].

Most cats (54%) diagnosed with SARS-CoV-2 infection were asymptomatic, and the virus could be identified only by molecular tests such as RT-qPCR, but not by clinical signs [28]. This finding is consistent with the findings in humans, wherein a high percentage of individuals positive for SARS-CoV-2 infection are asymptomatic [49]. Notably, the percentage of asymptomatic domestic cats is 54%, while that of asymptomatic humans is around 17%, varying between 4% and 52% [50]; however, it is probable that a high number of humans carrying the virus are not reported in the studies. This makes the association of asymptomatic individuals directly with diagnosed patients difficult, in contrast to the situation in cats. Moreover, the data on the percentage of asymptomatic humans are based on data from published papers and preprints. Consequently, important data generated by public health organizations in different countries, which are difficult to access and process, are not included [50]. Considering this, it is possible that the percentages of asymptomatic humans and cats are more similar than those reported.

Forty-six percent of the felines were diagnosed because they presented respiratory signs and needed veterinary care. The causative agent was subsequently identified by RT-qPCR [28]. Although symptomatic cats make up a significant percentage of the animals diagnosed with SARS-CoV-2 infection, to date, this infection in domestic cats is considered to be an infrequent condition with a mostly mild-to-moderate clinical course (Figure 4). Moreover, its prevalence is lower than the more “classical” respiratory infections caused by *Chlamydophila felis*, *Mycomplasma felis*, feline herpesvirus type-1 (FHV-1) and feline coronavirus (FCoV), which remain the main causes of respiratory problems [51].

Currently, SARS-CoV-2 is not the most prevalent causative agent of respiratory infections in the domestic feline population. However, considering the increasing reports of cats infected with the virus, it is possible that it will soon become an important agent in the veterinary landscape of respiratory infections in cats. This is probably due to the worldwide distribution of the virus and the exposure of cats through cohabitation with infected humans. Evidencing this, antibodies to SARS-CoV-2 have been detected in feline populations. For example, approximately 15% of cats in Wuhan, China evidenced immune response against SARS-CoV-2 [6]. In the city of Zaragoza, Spain, a smaller percentage (3.5%) of selected stray cats also showed antibodies against SARS-CoV-2 [52], suggesting a natural exposure to the virus. A recent study carried out from May to June 2020 in eastern France reported a seropositivity in 23.5% of the surveyed cats, as well as 15.3% of dogs [53]. Additionally, in Europe, a study surveyed cats and dogs in Italy for neutralizing antibodies against SARS-CoV-2 and reported 3.9% and 3.4%, respectively, of positive animals [54].

Although SARS-CoV-2 infections mostly affected respiratory tissues in cats, there is evidence of other clinical manifestations such as myocarditis, mainly related to infection with the British variant known as B.1.1.7. According to imaging studies, animals infected with this variant presented structural changes in the myocardium and complications such as ventricular arrhythmia and pleural and pulmonary effusion, without showing previous respiratory signs [31]. These findings highlight the multi-tropic capacity of SARS-CoV-2 and the relevance of assessing the impact of new variants on animal populations.

As previously described, in both humans and cats, the clinical course of SARS-CoV-2 infection is mainly respiratory, wherein symptoms such as coughing, lethargy and fever, in addition to digestive signs such as diarrhea, are common. The presentation of clinical signs common to humans and felids is biologically based on the presence and availability of the ACE2 receptor, which is used by SARS-CoV-2 to enter the host cells. This receptor is found in different tissues associated with the reported clinical signs in these species [11]. ACE2 is present in tissues of the lungs, liver, bladder, thyroid, heart, kidney, testis, small intestine and colon in humans [55]. In felines, ACE2 is present in the lungs, as well as in tissues of the skin, ears and retina [56]. It is therefore expected that one of the target tissues of SARS-CoV-2 in both humans and felids is the lung, which results in characteristic respiratory clinical manifestations that commonly occur in these species as well as other less common clinical signs in other tissues, which may also be associated with the presence of ACE2. As an example of this, ACE2 has been reported to be present in the gastrointestinal tract of cats and tigers [57], whereby the virus–receptor interaction and subsequent viral replication may occur in these tissues. This accounts for the viral RNA detected in fecal samples [12].

Considering that the intestine of cats and humans is rich in ACE2 receptors, fecal–oral transmission could be a route of transmission in these species [55,57]. Nevertheless, the chemical barrier posed by the stomach pH could hypothetically be a limiting factor for virus colonization in the gut, which may be one of the possible reasons for the lack of evidence of SARS-CoV-2 transmission from cats to humans.

Additionally, ACE2 shows high inter-species structural and amino acid homology. As demonstrated by in silico approaches, human ACE2 differs by only 14.2% from cat ACE2 and 13.8% from tiger ACE2. Moreover, at the interface between ACE2 and the receptor-binding domain of SARS-CoV-2 S protein, ACE2 sequences are almost identical in domestic cats and tigers and differ from the human sequence at only three residues. Two of the three variations correspond to the substitution of amino acids that belong to the same chemical group. The third variation involves the substitution of a neutral non-polar amino acid (methionine) for a neutral polar amino acid (threonine) [58]. These few differences between human and felid ACE2 could account for their susceptibility and similar clinical manifestations. It is worth noting that although domestic cats and tigers belong to different subfamilies (*Felinae* and *Pantherinae*, respectively), the amino acid sequence of the viral S protein-binding domain of ACE2 is identical in both. This could be extrapolated to other wild felines with SARS-CoV-2 infection, such as lion, snow leopard (*Pantherinae*) and puma (*Felinae*), for many of which molecular interaction studies cannot be conducted owing to the lack of information.

In addition to ACE2, transmembrane protease serine 2 (TMPRSS2) is also required for SARS-CoV-2 entry into the cell. TMPRSS2 plays a key role in ACE2 cleavage, which allows subsequent entry of the virus [59]. ACE2 and TMPRSS2 jointly determine the range of host cells that SARS-CoV-2 can enter. The results of in silico studies of interaction between the viral S protein and TMPRSS2 evince that species-specific mutations in TMPRSS2 (especially in mammals) do not affect the interaction between this receptor and the viral S protein [60].

Current reported cases of SARS-CoV-2 infection in domestic cats and wild felines appear to be linked to transmission events through direct contact with humans, owners or caretakers depending on the case [28]. Based on the above evidence regarding the homology of ACE2 orthologues in humans and felines, in addition to the similarity in tissues having high ACE2 expression, it would not be surprising that feline SARS-CoV-2 cases were the result of anthropozoonotic transmission. However, experimental infection studies on domestic cats have demonstrated animal–animal transmission [12], suggesting the potential for urban transmission among cat populations that arises from domestic cats allowed to leave the house.

Although cat–cat transmission occurs, SARS-CoV-2 infection in cats would be transient, and transmissibility would decrease over time. Bao et al. (2021) performed five virus passages between infected and healthy animals subsequently. Transmissibility decreased from one animal to the next and viral RNA was only detectable up to the second passage. This reduction in transmission would be related to the inherent characteristics of the feline virus-binding domain of the ACE2 receptor, which, compared to humans, has lower affinity to the virus [43].

Since cats are one of the species susceptible to symptomatic as well as asymptomatic SARS-CoV-2 infection, it is important to conduct further studies to elucidate the circulation and extent of the virus in cat populations associated with human populations, as domestic cats currently represent a high percentage of companion animals [61].

In addition, it is important to determine the potential these animals may have as reservoirs of the virus and the potential for reverse cat–human transmission. Likewise, the role these animals may play as possible independent foci of viral circulation, which may eventually lead to the recirculation of the virus to human populations, and its potential impact on public health need to be assessed. This is particularly important considering that virus circulation in animal populations can result in the emergence and establishment of mutant variants of the virus. Such is the case reported in mink farms in Denmark, where a new variant of the virus was detected both in minks and in some humans linked to these farms, resulting in the culling of the entire mink population in the country [62,63]. These mutations could represent future immune-escape variants, which can evade protection induced by vaccines recently made available to humans. In this context, it has recently been reported that within-host SARS-CoV-2 genetic variation is predominantly influenced by genetic drift and purifying selection. The transmission of SARS-CoV-2 between cats allows the selection of a specific viral variant at amino acid position 655 in virus Spike (H655Y) [64]. SARS-CoV-2 Spike mutant H655Y has been previously shown to confer escape from human monoclonal antibodies and is currently found in thousands of human sequences [65], suggesting that different species and context-specific adaptations are likely to continue to emerge in humans and in animals, underscoring the importance of continued genomic surveillance in humans and non-human mammalian hosts [64,66,67].

## 5. Conclusions

Given the current complex situation which has arisen due to the COVID-19 pandemic and the emergence of viruses with zoonotic potential, it is of paramount importance to establish both national and international surveillance systems to closely monitor the development and occurrence of SARS-CoV-2 and other agents of public health relevance in animals and its possible “spill over” to humans and the possible “spill-back”—causing public health concerns—highlighting that all efforts should be focused on the concept and implementation of “One Health/One Medicine” to understand how aspects involving the environment, animals, and humans could contribute substantially to the control of epidemics in the near future [1,68].

## Figures and Tables

**Figure 1 animals-11-02056-f001:**
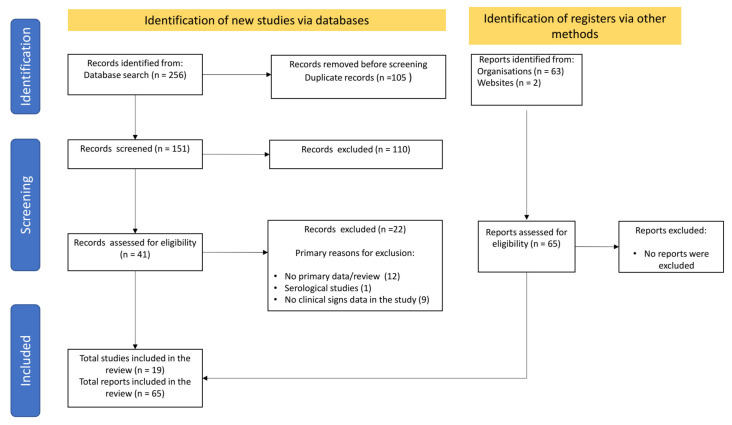
PRISMA workflow of search and selection of articles.

**Figure 2 animals-11-02056-f002:**
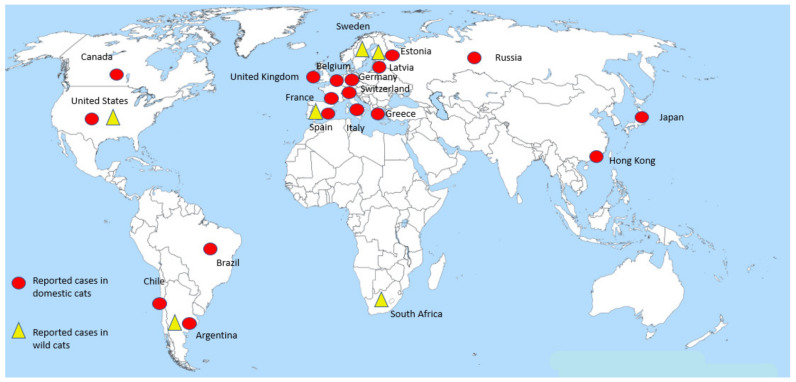
**Distribution of cases of SARS-CoV-2 reported in domestic and wild cats worldwide**. Red circles denote cases in domestic cats reported by country. The virus has been detected in at least three different continents (the Americas, Europe and Asia) in these animals. Yellow triangles denote cases in captive wild felines which have been reported in the Americas, Europe and Africa.

**Figure 3 animals-11-02056-f003:**
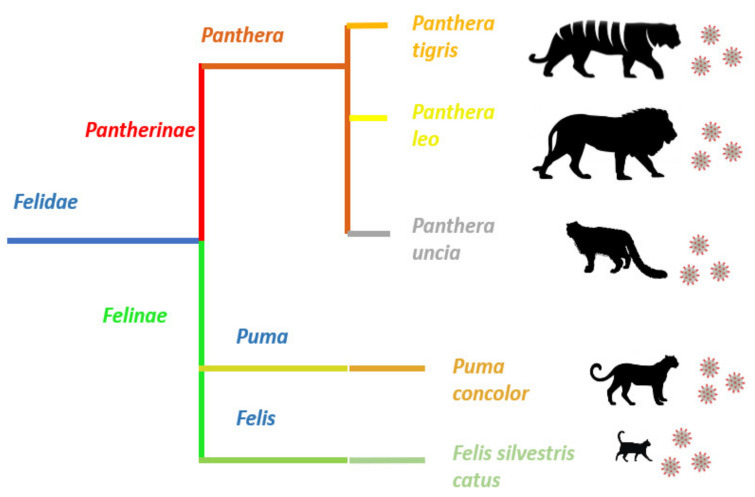
**Felid species reported with SARS-CoV-2 infection.** SARS-CoV-2 infection has been reported in the members of two subfamilies, *Pantherinae* and *Felinae*, belonging to the family *Felidae*. Despite the evident morphological differences between pantherids and felines, the virus is capable of infecting members of these two dissimilar families and causing very similar clinical signs.

**Figure 4 animals-11-02056-f004:**
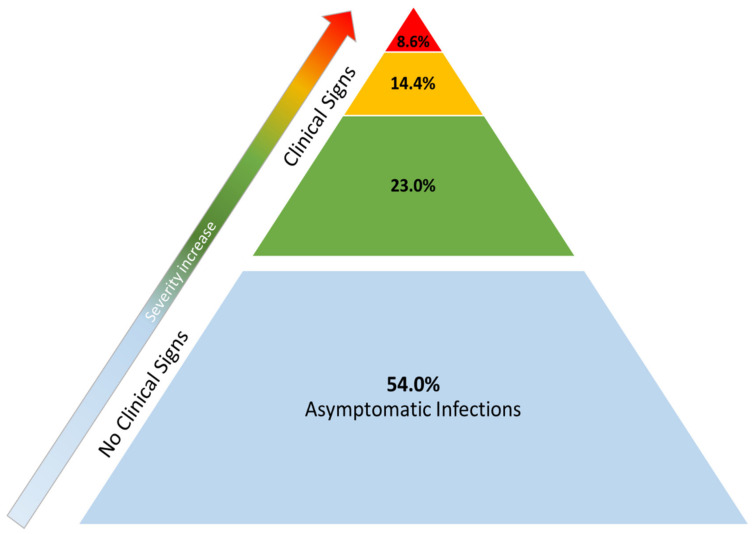
**Distribution of outcomes in cats with SARS-CoV-2 infection in published case reports**. The base of the pyramid denotes asymptomatic infections (blue area). In clinically ill animals, the vast majority of symptomatic cases presented mild clinical signs (green area). Approximately one third of the symptomatic animals showed medical complications of major clinical relevance (yellow area). A smaller percentage of animals (red area) eventually died after being diagnosed with SARS-CoV-2 infection: four of those had underlying health problems—which is why SARS-CoV-2 was considered incidental rather than a cause of death—and the remaining two were euthanized due to worsening of their clinical condition.

**Table 1 animals-11-02056-t001:** Clinical signs reported in domestic cats diagnosed with SARS-CoV-2 infection [27,29,31,33,34,35,36,37,38,39].

Clinical Signs and/or Findings	Number of Affected Animals	Percentage
Respiratory		
Sneezing	13	40.6%
Coughing	2	6.25%
Eye discharge	3	9.37%
Rales in lungs	2	6.25%
Dyspnea	4	12.51%
Tachypnea	2	6.25%
Congestion	1	3.1%
Nasal discharge	3	9.37%
Digestive		
Vomiting	1	3.1%
Diarrhea	2	6.25%
Mouth lesions	1	3.1%
Non-specific		
Fever	2	6.25%
Lethargy	12	37.5%
Lack of appetite	5	15.6%
Cardiovascular		
Hypertrophic cardiomyopathy	3	9.37%
Anemia	1	3.1%
Thrombocytopenia	1	3.1%
Congestive heart failure	8	25%
Ventricular arrhythmia	4	12.5%
Neurological		
Head pressing	1	3.1%

**Table 2 animals-11-02056-t002:** Clinical signs in wild felines resulting from SARS-CoV-2 infection [26,28,40,41].

Clinical Signs	Number of Affected Animals	Percentage
Coughing	28	96.5%
Sneezing	23	79%
Lack of appetite	15	51.7%
Nasal discharge	4	13.7%
Lethargy	4	13.7%
Wheezing	3	10.3%

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
