# Peer review of "SARS-CoV-2 Clinical Outcome in Domestic and Wild Cats: A Systematic Review"

_animals, 2021, doi:10.3390/ani11072056_

Round 1
Reviewer 1 Report
Simple summary and abstract need to be rephrased.
Line 76-- Add references
Line 84--Check the reference [19]
Line 141-Check the references [21,22], they don't seem to fit properly
Line 199_-Add reference
Line 203- Check the reference [27]
Line 415- I would suggest to add more SARS-CoV-2 serological survey on cats (Germany, Italy, Croatia, France,..)
Nice paper.
Author Response
REVIEWER 1
Simple summary and abstract need to be rephrased.
R/. DONE. We have rephrased some of them
Line 76-- Add references
R/. This line mentions the nature of the studies that are described later in the text. Each study was cited accordingly.
Line 84--Check the reference [19]
R/. We revised the cited article and confirmed that the citation is correct
Line 141-Check the references [21,22], they don't seem to fit properly
R/. We revised and modified the reference according to the recommendation.
Line 199_-Add reference
R/. Done
Line 203- Check the reference [27]
R/. We revised the cited article and confirmed that the citation is correct
Line 415- I would suggest to add more SARS-CoV-2 serological survey on cats (Germany, Italy, Croatia, France,..)
R/. We added additional surveys to this paragraph as suggested.
Nice paper.
R/. Thanks so much for your comments.
Reviewer 2 Report
In this manuscript, authors systematically summarize the clinical cases of SARS-CoV-2 infection in domestic cats and wild felids, which will be helpful for the prevention and control of SARS-CoV-2 in animals. The manuscript needs some editing as some sentence structures are awkward and some wording in inappropriate.
The authors summarize and present the clinical signs associated with SARS-CoV-2 infection in domestic cats and wild felids. At the beginning of the prevalence of COVID-19, domestic cats had been proven to be susceptible for SARS-CoV-2 via experimental infections. As the close partner in human life, the transmission of SARS-CoV-2 in domestic and wild cats could result in the emergence of virus mutants, which poses a risk to public health. Thus, this manuscript is timely and significative. This manuscript could provide detailed and scientific insights of the prevalence of SARS-CoV-2 in domestic and wild cats.
Major problems are:
- The prevalence of SARS-CoV-2 in domestic and wild cats, including nucleic acid detection by RT-PCR and antibody detection by ELISA, should be described in the manuscript. Actually, most cats infected with SARS-CoV-2 are asymptomatic, whether natural infections or experimental infections. The description of the prevalence will be helpful for the assessment of risk of SARS-CoV-2 infected in cats.
- In Fig. 4, the description of 18.75% death rate seems inappropriate, because most cats died due to other health problems, or not natural death.
Minor revisions are:
- Total 19 articles and 65 reports included in the review, but not all articles or reports are displayed in the main part and reference of this manuscript, why?
- Change Nidovirales, Coronaviridae, Orthocoronavirinae and Betacoronavirus to Nidovirales, Coronaviridae, Orthocoronavirinae and Betacoronavirus.
- Line 187 to 196, the logical relationship should be reorganized.
- In Table 1 and Table 2, the source of the clinical cases should be labelled as references.”
Author Response
Reviewer 2:
In this manuscript, authors systematically summarize the clinical cases of SARS-CoV-2 infection in domestic cats and wild felids, which will be helpful for the prevention and control of SARS-CoV-2 in animals. The manuscript needs some editing as some sentence structures are awkward and some wording in inappropriate.
The authors summarize and present the clinical signs associated with SARS-CoV-2 infection in domestic cats and wild felids. At the beginning of the prevalence of COVID-19, domestic cats had been proven to be susceptible for SARS-CoV-2 via experimental infections. As the close partner in human life, the transmission of SARS-CoV-2 in domestic and wild cats could result in the emergence of virus mutants, which poses a risk to public health. Thus, this manuscript is timely and significative. This manuscript could provide detailed and scientific insights of the prevalence of SARS-CoV-2 in domestic and wild cats.
Major problems are:
The prevalence of SARS-CoV-2 in domestic and wild cats, including nucleic acid detection by RT-PCR and antibody detection by ELISA, should be described in the manuscript. Actually, most cats infected with SARS-CoV-2 are asymptomatic, whether natural infections or experimental infections. The description of the prevalence will be helpful for the assessment of risk of SARS-CoV-2 infected in cats.
R/. This manuscript aimed to summarize que available information about the presence of SARS-CoV-2 in domestic and wild felids. During our review, we included the studies that reported positivity the virus or antibody detection in these animals. We did not include seroprevalence studies as those were not always statistically significant.
In Fig. 4, the description of 18.75% death rate seems inappropriate, because most cats died due to other health problems, or not natural death.
R/. We changed the word “Death” for “Deceased” and described the characteristics of these cases in the figure legend.
Minor revisions are:
Total 19 articles and 65 reports included in the review, but not all articles or reports are displayed in the main part and reference of this manuscript, why?
R/. As described in the Methods section, after the first search we performed a quality assessment which allowed us to filter the publications based on quality of data. We included the manuscripts that provided the best quality information possible for the review.
Change Nidovirales, Coronaviridae, Orthocoronavirinae and Betacoronavirus to Nidovirales, Coronaviridae, Orthocoronavirinae and Betacoronavirus.
R/. We changed the taxonomy to italics.
Line 187 to 196, the logical relationship should be reorganized.
R/. We agree to the reviewer and modified the text in order to be more readable.
In Table 1 and Table 2, the source of the clinical cases should be labelled as references.
R/. We included all references according to the Recommendations.
Reviewer 3 Report
SARS-CoV-2 infection in domestic and wild cats: A systematic review
The review by Giraldo-Ramirez presents an overview of the clinical symptoms observed in domestic and wild cats infected with SARS-CoV-2. The text reads well and an extensive review of the literature is proposed, together with the methodology to include or reject articles, which is much appreciated. This is a timely review since none focused on this specific aspect of an already neglected subject: SARS-CoV-2 infection in animals. Overall, the information presented here is clearly presented, although some redundancies can be noted and some transitions could be strengthened. My main concern is about the lack of caution or warning about the data available and the potential bias involved. Indeed, it is likely that there is a major bias toward the most severe clinical presentation, since only a minority of the cat are presented to a veterinarian with mild symptoms. Thus, asymptomatic cat detection can only occur in specific situations, resulting in a severe underestimation of asymptomatic and mild cases. In fact, seroprevalence studies indicate clearly that infection of cats in the natural condition is not unusual (especially in COVID-19 positive households), with prevalence ranging from 1-40% according to different studies (zhang et al - already cited in this paper but also fritz et al, stevanovic et al. and others…). I understand that the authors wanted to narrow the amount of data discussed here, but not considering seroprevalence studies to discuss clinical signs results in a biased message of the review, which is particularly visible in figure 4, where one could think that SARS-CoV-2 infection in cats is associated with almost 20% mortality (which I guess is not the message that the authors want to share, but the presentation is rather confusing). Hence, I strongly advise the authors to modify the manuscript accordingly, to highlight the fact that most of cats’ infections are likely to be asymptomatic and/or to add a warning at the beginning of the discussion pointing to these bias (to be fair, authors have already partially mentioned these weaknesses but to slightly in my opinion). However, apart from these main concerns, it is useful to summarize and describe clinical signs descriptions and frequencies, which is done well in this review.
In addition to the two main weaknesses highlighted above (to little warning of bias involved in clinical case reports and absence of seroprevalence studies not considered), I think that a figure summarizing the time-course of the infection (possibly with symptoms duration, and periods of virus excretion and/or detection would be very useful. Also, it is sometimes difficult to find the original paper from the text, maybe adding to table 1 the ref where animals are described would be very useful as well. In addition, it is said in the mat and meth section that 65 + 19 reports are included, but there are only 58 references in the article. I think that all papers used for the review should be referenced or discard from the review. I would also prefer (but this is only my opinion and other presentation could be efficient as well) to have the lab experiments data presented in the first position (since they are likely to be more accurate) and in second position natural infection to discuss how SARS-CoV-2 infection presents in “real” life.
Some references are prepublication (biorxiv), it should be mentioned when such articles are cited (now only written line 103 in the mat and meth section). Also, some references are inappropriately cited, such as ref 1 line 44 or ref 21 and 22 which refers to avian influenza and canine distemper virus infections, with little relevance to the subject (only that they are papers from some co-authors).
Line 24-26: articles, detailed and press reports are were analyzed and only articles and “general” reports were included. What are the general reports and how different are they from detailed reports?
Line 42-44: I don’t see the logic between the two sentences, it is not because coronaviruses are infecting different species that they are necessarily jumping the species barrier easily.
Line 31-33: The sentence message is unclear, please explain
Line 47-48: I don’t agree with the statement, most of the work has focused on treatment or immune response, little has been done on the origin for instance.
Line 52 to 53 and rest of the paper: groups (order, family, genus etc…) should be written in italic
Line 54-56: The intermediate host not found already, hosts that is indicated here (snake, pangolins etc…) have been proven not to be the intermediate hosts.
Line 60-65: the role of ACE-2 in the susceptibility of the species is probably more complex and other factors (such as restriction factors) are probably playing a role (ex, some animal species have lower ACE-2 homology but shows higher susceptibility to SARS-CoV-2)
Line 76: mainly? I don’t think other situations have been described in natural conditions. If not please elaborate.
Line 84: I would be more cautious here since only a few articles analyzed such possibilities
Line 140-141: the articles cited here are unrelated to the present study and should be removed
Line 154: 65 instead of 55?
Line 170 and 171: It also due to the higher level of vet medicine in these areas otherwise, high prevalent countries such as Brazil or Mexico would also be present?
Line 183 : ref 21 is unrelated, ref to all the papers cited here (in the list of countries) should appear or should be summarized in table 1 for example
Line 225: more variants description could be introduced here and then the transition “more specifically” would be appropriate to introduce the English variant
Line 233-234: it is rather impossible to precisely determine the infection date in a cat in natural condition, maybe you could add a warning here?
Line 216 to 224 are redundant with line 231 to 246
Figure 3: What is the meaning of the colors? what does the virus cartoon means? Have other felids been shown to be resistant to SARS-CoV-2? Maybe adding more felids species in the tree and highlighting those that have been detected as susceptible would be useful, but most of the felids may be susceptible to SARS-CoV-2, so I wonder if this figure is useful?
Line 320-321: Has another situation been demonstrated?
Line 375-376: cat-cat transmission is described in the lab experiment but has never been demonstrated in natural conditions!
Line 381-390: I don’t think that % of asymptomatic infection in humans is such a mystery. For cats, prevalence studies indicate very clearly that cat infection is common in a natural condition (especially in SARS-CoV-2 positive households with 10-40% of cat positives for SARS-CoV-2 and a minority being symptomatic)
Line 396: I don’t know how SARS-CoV-2 is considered but this statement should be strengthened by a ref
Line 398: the study was conducted in early 2020, some data about the prevalence would be useful, maybe the main difference is that most of the diseases have a vaccine?
Figure 4: the legend title is inaccurate since most of the infected cats are asymptomatics
Line 460: ad space between “TMPRSS2” and “plays”
Line 466: are there other cases described?
Author Response
The review by Giraldo-Ramirez presents an overview of the clinical symptoms observed in domestic and wild cats infected with SARS-CoV-2. The text reads well and an extensive review of the literature is proposed, together with the methodology to include or reject articles, which is much appreciated. This is a timely review since none focused on this specific aspect of an already neglected subject: SARS-CoV-2 infection in animals. Overall, the information presented here is clearly presented, although some redundancies can be noted and some transitions could be strengthened. My main concern is about the lack of caution or warning about the data available and the potential bias involved. Indeed, it is likely that there is a major bias toward the most severe clinical presentation, since only a minority of the cat are presented to a veterinarian with mild symptoms. Thus, asymptomatic cat detection can only occur in specific situations, resulting in a severe underestimation of asymptomatic and mild cases. In fact, seroprevalence studies indicate clearly that infection of cats in the natural condition is not unusual (especially in COVID-19 positive households), with prevalence ranging from 1-40% according to different studies (zhang et al - already cited in this paper but also fritz et al, stevanovic et al. and others…). I understand that the authors wanted to narrow the amount of data discussed here, but not considering seroprevalence studies to discuss clinical signs results in a biased message of the review, which is particularly visible in figure 4, where one could think that SARS-CoV-2 infection in cats is associated with almost 20% mortality (which I guess is not the message that the authors want to share, but the presentation is rather confusing).
R/. We totally agree to the reviewer and understood the difficulties presented by the reviewer. Multiple paragraphs had been reorganized and text modified to a better presentation of the paper including a modification in the title of the paper to focus on the clinical picture of the infected animals. Also figure 4 was updated to accomplish this Aim.
Hence, I strongly advise the authors to modify the manuscript accordingly, to highlight the fact that most of cats’ infections are likely to be asymptomatic and/or to add a warning at the beginning of the discussion pointing to these bias (to be fair, authors have already partially mentioned these weaknesses but to slightly in my opinion). However, apart from these main concerns, it is useful to summarize and describe clinical signs descriptions and frequencies, which is done well in this review.
R/. According the reviewer recommendation, we included a text at the Weakness section and move it before Discussion to clarify the proper bias from this kind of analysis. Also this were included y the Abstract of the paper.
In addition to the two main weaknesses highlighted above (to little warning of bias involved in clinical case reports and absence of seroprevalence studies not considered), I think that a figure summarizing the time-course of the infection (possibly with symptoms duration, and periods of virus excretion and/or detection would be very useful.
R/. We agree to the reviewer. However, although we tried to develop a figure around the time-course of the infection, the lack of sufficient information did not allow us to achieve it. Mainly because in none of the cases is the date of infection of the animals certain and there is only an association with the onset of the signs and a very long recovery time after the onset of the signs (7-15 days) ( Lines 265-269)
Also, it is sometimes difficult to find the original paper from the text, maybe adding to table 1 the ref where animals are described would be very useful as well. In addition, it is said in the mat and meth section that 65 + 19 reports are included, but there are only 58 references in the article. I think that all papers used for the review should be referenced or discard from the review. I would also prefer (but this is only my opinion and other presentation could be efficient as well) to have the lab experiments data presented in the first position (since they are likely to be more accurate) and in second position natural infection to discuss how SARS-CoV-2 infection presents in “real” life.
R/. We agree to the reviewer. That why most USDA reports were discarded from the systematic analysis. A List of references used for the Tables 1 and 2 were included according to the reviewers recommendations.
Some references are prepublication (biorxiv), it should be mentioned when such articles are cited (now only written line 103 in the mat and meth section). Also, some references are inappropriately cited, such as ref 1 line 44 or ref 21 and 22 which refers to avian influenza and canine distemper virus infections, with little relevance to the subject (only that they are papers from some co-authors).
R/. We agree to the reviewer. References were modified. By involuntary error, we mistaken sone of the references of previous works in our laboratory. Most of BioRxiv references were updated to the current full papers in journals.
Line 24-26: articles, detailed and press reports are were analyzed and only articles and “general” reports were included. What are the general reports and how different are they from detailed reports?
R/. We corrected those typo mistakes in order for a better readability of the paper.
Line 42-44: I don’t see the logic between the two sentences, it is not because coronaviruses are infecting different species that they are necessarily jumping the species barrier easily.
R/. This was corrected to improve the logic in the sentence.
Line 31-33: The sentence message is unclear, please explain
R/. This was corrected.
Line 47-48: I don’t agree with the statement, most of the work has focused on treatment or immune response, little has been done on the origin for instance.
R/. We understand that there is a significant number of studies focused on prevention and treatment of COVID-19, in comparison with basic studies. However, there is also several studies addressing the origin of the virus. In fact, one of our authors published a manuscript were we studied this topic from the phylogenetic point of view, citing several studies in the same topic.
Line 52 to 53 and rest of the paper: groups (order, family, genus etc…) should be written in italic
R/. This was properly corrected.
Line 54-56: The intermediate host not found already, hosts that is indicated here (snake, pangolins etc…) have been proven not to be the intermediate hosts.
R/. This was properly corrected.
Line 60-65: the role of ACE-2 in the susceptibility of the species is probably more complex and other factors (such as restriction factors) are probably playing a role (ex, some animal species have lower ACE-2 homology but shows higher susceptibility to SARS-CoV-2)
We agree with this statement. However, we think it is important to highlight the role of ACE2 in other species, as we have discussed this in previous publications, and it is still considered the major element for determining the susceptibility to SARS-CoV-2.
Line 76: mainly? I don’t think other situations have been described in natural conditions. If not please elaborate.
R/. We strongly agree to the reviewer. This was properly corrected.
Line 84: I would be more cautious here since only a few articles analyzed such possibilities
This was properly corrected.
Line 140-141: the articles cited here are unrelated to the present study and should be removed.
R/. We strongly agree to the reviewer. This was properly corrected.
Line 154: 65 instead of 55?
R/. We apologize for this typo mistake. This was properly corrected.
Line 170 and 171: It also due to the higher level of vet medicine in these areas otherwise, high prevalent countries such as Brazil or Mexico would also be present?
We agree and included a sentence in the manuscript.
Line 183 : ref 21 is unrelated, ref to all the papers cited here (in the list of countries) should appear or should be summarized in table 1 for example
R/. We apologize for this typo mistake. This was properly corrected.
Line 225: more variants description could be introduced here and then the transition “more specifically” would be appropriate to introduce the English variant
R/. We agree that new variants could change the current view of the SARS-CoV-2 infections in domestic and wild felids, as they could introduce lower/higher affinity for infection in these species. However, there is not available information about other variants and felids yet.
Line 233-234: it is rather impossible to precisely determine the infection date in a cat in natural condition, maybe you could add a warning here?
R/. We corrected this.
Line 216 to 224 are redundant with line 231 to 246
R/. We agree this could sound redundant. However, the second paragraph presented a more detailed description of the frequency of the clinical signs reported previously. We corrected the wording to better connect the two sections.
Figure 3: What is the meaning of the colors? what does the virus cartoon means? Have other felids been shown to be resistant to SARS-CoV-2? Maybe adding more felids species in the tree and highlighting those that have been detected as susceptible would be useful, but most of the felids may be susceptible to SARS-CoV-2, so I wonder if this figure is useful?
R/. We partially agree to the reviewer. Although it has been previously established and reported that most felines may have biological susceptibility to SARS-CoV-2 (Damas et al., 10.1073/pnas.2010146117), the purpose of the representation is to present the differences at the subfamily level within the Felidae family in the confirmed SARS-CoV-2 infected animal.
Line 320-321: Has another situation been demonstrated?
R/. We modify the text for a better readability
Line 375-376: cat-cat transmission is described in the lab experiment but has never been demonstrated in natural conditions!
R/. We agree, so we did not state cat-cat natural transmission is happening. However, based on the experimental data, we think it is safe to suggest natural transmission could be possible.
Line 381-390: I don’t think that % of asymptomatic infection in humans is such a mystery. For cats, prevalence studies indicate very clearly that cat infection is common in a natural condition (especially in SARS-CoV-2 positive households with 10-40% of cat positives for SARS-CoV-2 and a minority being symptomatic)
R/. As you said, asymptomatic infections in humans it is not a mystery and we aware are of. This sentenced aimed to highlight what could be happening in domestic cats.
Line 396: I don’t know how SARS-CoV-2 is considered but this statement should be strengthened by a ref
R/. This was corrected accordingly.
Line 398: the study was conducted in early 2020, some data about the prevalence would be useful, maybe the main difference is that most of the diseases have a vaccine?
R/. This was corrected accordingly.
Figure 4: the legend title is inaccurate since most of the infected cats are asymptomatics
R/. This was corrected accordingly.
Line 460: ad space between “TMPRSS2” and “plays”
R/. This was corrected accordingly.
Line 466: are there other cases described?
R/. This was corrected accordingly.
Round 2
Reviewer 3 Report
Most of the comments were answered accordingly to the recommendation and the overall quality of the manuscript has increased since the previous version.
However, the current version retains two weak points that should be fixed before publication. Once corrected, I think this review can be published without further modifications.
- I still have an issue with figure 4 which in my opinion provide a misleading message. I understand that it is not the intention of the authors and that they are perfectly aware that most of the infection are asymptomatic in cats, but at first glance and with the title “Distribution of outcomes of SARS-CoV-2 infection in domestic cats”, this figure seems to indicate that almost 8.6% of infection ultimately end in the death of the infected animal. The legend clarifies this point but I am still concerned that this figure could mislead the reader. I suggest removing figure 4 in the final version of the article. Alternatively, modify the title with “Distribution of outcomes in cats with SARS-CoV-2 infection in published case reports” or something similar.
- Authors indicate that 65 reports and 19 articles are included in the review, however, there are only 67 references. My position is that all the articles included in any review should be found in the references.
Other minor corrections :
Line 43 : Typo (a dot should be removed)
Line 82 : Typo (a dot is missing)
Line 169-170 : check the sentence
Author Response
Thank you for your comments. We addressed the typos and corrected the suggested sentence. About the figure 4, we changed the figure legend title, according to your suggestion. We think that this figure provides a general view of the distribution of the disease’s outcomes, based on the available reports. We stated the 8.6% corresponded to clinical cases that were diagnosed with SARS-CoV-2 but the virus infection was not confirmed as the cause of dead.
We want to clarify, as we mentioned in the method section, we filtered these manuscripts and removed reviews, seroprevalence studies, cases in non-feline species and reports related to infections in humans or vaccination. After this filtering and after revising the quality and relevance of their content, we went down to 19 manuscripts that were included in our review. The "reports" are synthetized in the OIE and USDA references and cannot be cited individually.